Leveraging machine learning to uncover multi-pathogen infection dynamics across co-distributed frog families

http://orcid.org/0000-0003-0042-0825 Wiley Daniele L. F. 1 dwiley7@unm.edu
Omlor Kadie N. 1
Torres López Ariadna S. 1
Eberle Celina M. 1
Savage Anna E. 2
Atkinson Matthew S. 2
http://orcid.org/0000-0001-7081-2432 Barrow Lisa N. 1
1 Museum of Southwestern Biology, Department of Biology, University of New Mexico , Albuquerque, New Mexico , United States
2 Department of Biology, University of Central Florida , Orlando, Florida , United States
Hughes Daniel
Electronic publication date: 2025 Jan 29
Publication date: 2025
Volume: 13
Electronic Location ID: e18901
Received 2024 Oct 18; Accepted 2025 Jan 3
Copyright: © 2025 Wiley et al.
Copyright year: 2025
Copyright holder: Wiley et al.
License: This is an open access article distributed under the terms of the Creative Commons Attribution License, which permits unrestricted use, distribution, reproduction and adaptation in any medium and for any purpose provided that it is properly attributed. For attribution, the original author(s), title, publication source (PeerJ) and either DOI or URL of the article must be cited.
License URL: https://creativecommons.org/licenses/by/4.0/

Keywords: Batrachochytrium dendrobatidis, Ranavirus, Amphibian Perkinsea, Random forests, Amphibian disease, Bufonidae, Hylidae, Ranidae

Funding: National Science Foundation Graduate Research Fellowship Program 2439853 NSF DEB-2112946 University of New Mexico (UNM) Office of the Vice President for Research through a Research Allocations Committee award This material is based upon work supported by the National Science Foundation Graduate Research Fellowship Program to Daniele L. F. Wiley under Grant No. (2439853) and funding to Lisa N. Barrow from NSF (DEB-2112946) and the University of New Mexico (UNM) Office of the Vice President for Research through a Research Allocations Committee award. Any opinions, findings, and conclusions or recommendations expressed in this material are those of the author(s) and do not necessarily reflect the views of the funders. The funders had no role in study design, data collection and analysis, decision to publish, or preparation of the manuscript.

==============================
Background

Amphibians are experiencing substantial declines attributed to emerging pathogens. Efforts to understand what drives patterns of pathogen prevalence and differential responses among species are challenging because numerous factors related to the host, pathogen, and their shared environment can influence infection dynamics. Furthermore, sampling across broad taxonomic and geographic scales to evaluate these factors poses logistical challenges, and interpreting the roles of multiple potentially correlated variables is difficult with traditional statistical approaches. In this study, we leverage frozen tissues stored in natural history collections and machine learning techniques to characterize infection dynamics of three generalist pathogens known to cause mortality in frogs.

Methods

We selected 12 widespread and abundant focal taxa within three ecologically distinct, co-distributed host families (Bufonidae, Hylidae, and Ranidae) and sampled them across the eastern two-thirds of the United States of America. We screened and quantified infection loads via quantitative PCR for three major pathogens: the fungal pathogen Batrachochytrium dendrobatidis (Bd), double-stranded viruses in the lineage Ranavirus (Rv), and the alveolate parasite currently referred to as Amphibian Perkinsea (Pr). We then built balanced random forests (RF) models to predict infection status and intensity based on host taxonomy, age, sex, geography, and environmental variables and to assess relative variable importance across pathogens. Lastly, we used one-way analyses to determine directional relationships and significance of identified predictors.

Results

We found approximately 20% of individuals were infected with at least one pathogen (231 single infections and 25 coinfections). The most prevalent pathogen across all taxonomic groups was Bd (16.9%; 95% CI [14.9–19%]), followed by Rv (4.38%; 95% CI [3.35–5.7%]) and Pr (1.06%; 95% CI [0.618–1.82%]). The highest prevalence and intensity were found in the family Ranidae, which represented 74.3% of all infections, including the majority of Rv infection points, and had significantly higher Bd intensities compared to Bufonidae and Hylidae. Host species and environmental variables related to temperature were key predictors identified in RF models, with differences in importance among pathogens and host families. For Bd and Rv, infected individuals were associated with higher latitudes and cooler, more stable temperatures, while Pr showed trends in the opposite direction. We found no significant differences between sexes, but juvenile frogs had higher Rv prevalence and Bd infection intensity compared to adults. Overall, our study highlights the use of machine learning techniques and a broad sampling strategy for identifying important factors related to infection in multi-host, multi-pathogen systems.

Introduction

In the last century, amphibians have experienced declines and extinctions attributed to a myriad of anthropogenic stressors, including climate change, habitat destruction, and species introductions (Bellard, Genovesi & Jeschke, 2016; Miller et al., 2018; Cordier et al., 2021). These factors have collectively led to classifying 41% of known species as threatened or endangered by the International Union for Conservation of Nature (Luedtke et al., 2023). Investigations following enigmatic declines have further identified the emergence of infectious diseases as a significant contributor to amphibian biodiversity loss (Berger et al., 1998; Smith, Acevedo-Whitehouse & Pedersen, 2009; Scheele et al., 2019; Rollins-Smith, 2020). With the projected increase in infectious disease spread alongside anticipated climatic shifts (Rollins-Smith, 2017; Price et al., 2019), there is a pressing need for continued disease surveillance and assessment of multi-pathogen infection dynamics within host communities.

Across North American frogs, mortality events have been linked to three major pathogens—the aquatic fungus Batrachochytrium dendrobatidis (Bd, Scheele et al., 2019), double-stranded viruses in the genus Ranavirus (Rv, Miller, Gray & Storfer, 2011), and the protozoan endoparasite known as Amphibian Perkinsea (Pr, Isidoro-Ayza et al., 2017). Since their respective discoveries, research has shown that infection prevalence and impact vary across and within host taxa (e.g., Greenberg, Palen & Mooers, 2017), demographic traits such as age (Humphries et al., 2022) and sex (Adams et al., 2017; Belasen et al., 2019), and environmental conditions related to latitude, elevation, and seasonality (Petersen et al., 2016; Whitfield et al., 2017; Sasso, McCallum & Grogan, 2021). These studies, however, are often limited by reduced taxonomic and geographic breadth and investigate only one pathogen at a time. Standardized screening efforts for multiple pathogens across susceptible host species sampled within the same environments are needed to effectively capture the complex interactions of factors driving infectious disease patterns.

Although Bd, Pr, and Rv can infect a diverse range of host taxa, susceptibility is not uniform across all taxonomic groups (Bancroft et al., 2011) and certain life history traits have been attributed to pathogen-specific infection risk. For example, due to the reliance on water as a mechanism for pathogen spread and persistence, Bd, Pr, and Rv infections are more prevalent in species associated with aquatic habitats (Bd—Greenberg, Palen & Mooers, 2017), such as ephemeral ponds (Rv—Hoverman et al., 2011; Pr—Hartmann et al., 2024). Additionally, variation in infection outcomes across and within species has also been demonstrated both in ex situ pathogen challenge experiments (e.g., Bd: Gervasi et al., 2013; Becker et al., 2014; Rv: Hoverman, Gray & Miller, 2010) and in situ community-wide surveys (e.g., Bd: Whitfield et al., 2017). These differences in infection outcome among species may be explained by a combination of ecological and life history traits (Becker et al., 2014), host genetics (Savage, Becker & Zamudio, 2015), immunogenetics (Savage et al., 2019; Trujillo et al., 2021), as well as environmental factors known to impact the host’s immune responses (Raffel et al., 2006; Rollins-Smith, 2017), further highlighting the challenges of assessing broad infection dynamics in natural systems.

Additionally, host demographic traits such as age class (reviewed in Humphries et al., 2022) and sex (Adams et al., 2017; Belasen et al., 2019), as well as coinfection with multiple pathogens (Ramsay & Rohr, 2021; Atkinson & Savage, 2023b), can further explain population-level variation in infection dynamics. For example, some studies show that metamorphic and post-metamorphic adult frogs suffer higher mortality rates compared to larvae when infected with Bd (Abu Bakar et al., 2016; Humphries et al., 2024), presumably due to the higher percentage of keratinized skin cells which the fungus uses as its primary resource. In contrast, frogs during their adult life stage tend to have the lowest prevalence of Pr and Rv infection, likely due to the increased competence of the host’s immune system after metamorphosis (Gray, Miller & Hoverman, 2009; Karwacki et al., 2018; Karwacki, Martin & Savage, 2021). Moreover, in some species, males exhibit higher Bd prevalence (Adams et al., 2017) and lower Bd survival rates compared to females (Carey et al., 2006), which could be because of differences in behavior related to exposure (e.g., males congregate in water for longer periods during breeding) or physiology (e.g., higher levels of testosterone may suppress immune function). To complicate things even further, harboring multiple pathogens can impact host tolerance to infection (Johnson & Buller, 2011), with coinfections of Bd and Rv resulting in inhibited growth and increased mortality (Ramsay & Rohr, 2021). Due to the context-dependent nature of interactions between host and pathogen dynamics, capturing this broad variation in susceptibility and mortality across communities is challenging and requires extensive surveys to better understand the generality of these patterns.

Environmental variables, including aspects of temperature and precipitation, influence pathogen distributions across susceptible hosts (e.g., Sasso, McCallum & Grogan, 2021). Though Bd and Rv differ in their thermal limits (Longcore, Pessier & Nichols, 1999; Ariel et al., 2009), with Bd being more sensitive to higher temperatures, both pathogens have been reported across much of North America (Peterson & McKenzie, 2014; Bartlett et al., 2021). Pathogen mediated mortality, however, has been strongly associated with temperature seasonality, where Bd and Rv die-off events tend to peak over winter months, at more northern latitudes, and at higher elevations (Bd: Raffel et al., 2013, 2015; Li et al., 2021; Rv: Gray, Miller & Hoverman, 2009; Balseiro et al., 2010; Rollins-Smith, 2017). Little is currently known about the full distribution and limits of Pr, but seasonal outbreaks have been reported in the Southeastern U.S., mirroring conditions related to Bd and Rv mortality events in this region (Karwacki et al., 2018; Atkinson & Savage, 2023b). The overlap in geographic distribution, suitable environmental conditions, and susceptible host species across all three pathogens leads to high potential for pathogen interactions and warrants further study.

One reason for this gap in research is that multi-host, multi-pathogen studies face sampling challenges due to the number of specimens required to detect pathogens and derive meaningful disease patterns (Stallknecht, 2007). Despite targeted research efforts, achieving sufficient taxonomic and geographic coverage is logistically difficult in a single study. Additionally, while less invasive methods like skin swabs are useful in Bd monitoring surveys, particularly of sensitive species, they are often single-use and may be less accurate at detecting low-level infections and other pathogens (Bd: DiRenzo & Campbell Grant, 2019; Rv: Miller, Gray & Storfer, 2011). Natural history collections offer a potential solution and opportunity for wildlife disease studies (Colella et al., 2021; Thompson et al., 2021). Linking pathogen and parasite data with a vouchered host specimen aligns with the FAIR Data Principles (defined in Wilkinson et al., 2016), which aim to increase data Findability, Accessibility, Interoperability, and Reusability. Vouchered samples facilitate extended genomic, physical, and molecular analyses critical to key discoveries in disease studies (Colella et al., 2021; Karwacki, Martin & Savage, 2021). Therefore, integrating natural history collections into the study of amphibian diseases can help address the large-scale data requirements needed to understand multi-pathogen dynamics, while also allowing for future research to more easily expand on previous discoveries.

Even with adequate sampling, disentangling the interconnected traits related to host-pathogen dynamics is challenging with standard linear statistics (O’Brien, Van Riper & Myers, 2009). Given the high correlation and nonlinearity of potential predictor variables (e.g., aspects of geography, temperature, precipitation), one flexible tool to analyze these complex factors is machine learning. Unlike traditional linear models, machine learning is adept at managing large datasets with numerous correlated variables and is frequently employed in bioinformatic applications (Schwalbe & Wahl, 2020). Specifically, random forests (RF) is a machine learning algorithm that integrates multiple decision trees to predict a response based on many selected variables (Breiman, 2001). This approach can identify features important to amphibian host-pathogen dynamics (Bancroft et al., 2011); however, RF is highly sensitive to class imbalance, which is common when handling disease datasets where one response (e.g., either infected or uninfected) is more strongly represented than the other. In such cases, models are built for and prioritize prediction accuracy of majority cases (e.g., uninfected status) while overlooking the minority cases (e.g., infected status). To address this issue, balanced RF techniques, such as building models with random equal sampling of both majority and minority cases through either downsampling the majority class or upsampling the minority class, can be employed as a correction (Chen, Liaw & Breiman, 2004). In this study, we use balanced RF to evaluate the relative importance of host taxonomy, demographic traits, and geographic and environmental factors for predicting pathogen-specific infection status and intensity.

Investigating geographic regions and species with high pathogen occurrence, but relatively low population decline, can provide valuable insights about the implied infection risk of sensitive co-occurring species. Here, we used range-wide comparisons of widespread and abundant frog species sampled across the central and eastern U.S. to characterize infection status and intensity across multiple hosts, pathogens, and environments. First, we described the prevalence and intensity of Bd, Pr, and Rv infections and coinfections within three anuran families using museum tissue collections, allowing the assessment of differences across host taxonomy and geographic distributions. Second, we applied machine learning to determine which host traits and environmental factors were most important for predicting pathogen infection status. Lastly, we validated our models and further investigated correlation and directional relationship of identified predictors, i.e., host family, species, age, sex, latitude, elevation, and associated temperature and precipitation variables related to pathogen occurrence and infection loads using traditional statistical methods.

Materials and Methods

Sampling

We obtained tissue samples through field collections and museum loans (Table S1). Individuals were largely sampled during summer breeding months from 2009–2023 under appropriate state and local permits (see Field Study Permissions below) and were archived at the Museum of Southwestern Biology (MSB), University of New Mexico (UNM). Post-metamorphic frogs were captured by hand, often along public roadways or near lentic waterways, while tadpoles were collected via dipnet. Individuals were either toe clipped and released at the site of capture or euthanized shortly after capture following protocols approved by the UNM Institutional Animal Care and Use Committee (Protocols 20-201006-MC & 23-201375-MC) and Florida State University (FSU) Animal Care and Use Committee (Protocols 1017 & 1313). Specifically, we applied 20% benzocaine to the ventral side of the frog, as described in the Guidelines for Use of Live Amphibians and Reptiles in Field and Laboratory Research (Herpetological Animal Care and Use Committee (HACC), 2004). Once the frog was completely unresponsive, we removed the heart as a secondary means of euthanasia. Dissection tools were flame sterilized between individuals, but not between tissue types collected. Tissues (toe, tail, muscle, and/or liver) were preserved either in 95% ethanol or tissue buffer (20% DMSO, 0.25 M EDTA, salt-saturated), or were immediately flash frozen in liquid nitrogen and kept frozen until DNA extraction.

In total, we obtained 1,281 samples from 32 states across the central and eastern U.S. (Fig. 1). Samples were selected to represent three major anuran families (Bufonidae n = 320, Hylidae n = 456, Ranidae n = 505) with four taxa in each family (11 species and one species complex; Fig. 1). Specifically, we targeted common and abundant species with overlapping distributions to facilitate comparisons across families while controlling for geographic and environmental variables.

Figure 1 Maps of sampling distribution by taxonomic group.

(A–C) Aggregated sampling depth for each family, with the number of individuals sampled shown at each site. (D–O) Species within each family sampled. Black dots represent sampled sites, and the colored regions depict native ranges obtained from the international union for conservation of nature red list (IUCN, 2022). (J) Combined ranges of cryptic species Hyla chrysoscelis and H. versicolor. Silhouettes of frogs were obtained from phylopic.org.

Molecular methods

Genomic DNA was extracted from tissue samples that consisted of either external only (toe, tail n = 259), internal only (liver, muscle n = 363), or a combination of both tissue types (n = 586; Fig. S1). We note that the use of internal tissues is not recommended for detecting Bd (World Organisation for Animal Health (WOAH), 2021a); however, during dissection, Bd zoospores are occasionally transferred from the skin to internal tissues. We therefore expected occasional, but not reliable detection of Bd from internal tissues (demonstrated in a comparison by Torres López et al., 2024), which is why extractions of these samples were included in select downstream analyses.

We used the E.Z.N.A. Tissue DNA Extraction Kit (Omega Bio-tek, Inc, Norcross, GA, USA) following the manufacturer’s recommendations. Whole genomic DNA concentrations were determined using the Broad Range Qubit Assay (Invitrogen, Eugene, OR, USA) and standardized to DNA concentrations of 9–15 ng/µL. Pathogen presence and quantity were determined via quantitative PCR (qPCR) following established protocols (Bd—Boyle et al., 2004; Rv—Allender, Bunick & Mitchell, 2013; Pr—Karwacki et al., 2018). Reactions were prepared in a UV-sterilized, clean air workstation. For each pathogen-specific qPCR, we used 25 μL reaction volumes, each containing 5 μL of extracted DNA template, 2 µL each of 10 μM forward and reverse primers, 5 µL of 1 μM probe (Eurofins Genomics, Louisville, KY, USA), and 8 µL SsoAdvanced Universal Probes Supermix (Bio-Rad, Hercules, CA, USA). Serial dilutions of pathogen-specific gBlocks (idtDNA, Coralville, IA, USA) were mixed with 0.1 ng/μL yeast tRNA carrier (Eurofins Genomics) to prevent small nucleic fragments from binding to the low-bind tubes and were used in duplicate as standards on each plate. Two positive controls (5 μL at 10 ng/μL concentration of known positive samples: Bd—MSB:Herp:104601; Rv—MSB:Herp:104600; Pr—MSB:Herp:104643) and two negative controls (molecular grade water) were included on each plate. Plates were run on a CFX96 Touch Real-Time PCR Detection System (Bio-Rad, Hercules, CA, USA), with cycling conditions of 95 °C for 5 min, followed by 40 cycles of 95 °C for 15 s and either 60 °C for 1 min (Bd/Rv) or 59 °C for 1 min (Pr).

Initial screening included two independent runs on each sample for each pathogen type; samples that tested negative in both runs were deemed uninfected, requiring no further screening. Partial and consistent positives underwent re-testing on two more independent plates to rule out false positives and to obtain final infection intensity values, resulting in at least four independent qPCR runs. Infection intensity values were calculated using the amplification curves (Cq) converted into starting quantities of pathogen DNA (SQ), with averages derived from the final two qPCR runs to account for differences in reagent sensitivity between runs. Individuals testing positive half of the time over the four plates of screening were categorized as “low” infections and included in all prevalence evaluations, but their infection intensity values were unreliable and were not considered in downstream analyses.

Calculating pathogen prevalence and infection loads

Pathogen prevalence was determined by summing the number of individuals found positive for each pathogen within a family, species, age, and sex, divided by the total number of samples within that category (Tables S2–S4). Because of the relatively low proportion of infected individuals within groups, we calculated 95% binomial confidence intervals using the logit method available in the binom package in R (R Core Team, 2023; Sundar, 2006). All individuals were taxonomically assigned and identified to the species or species complex level (i.e., Hyla chrysoscelis-versicolor). Samples lacking age class or sex identification at the time of collection were categorized as an unknown and were excluded from prevalence or infection intensity comparisons for those factors.

Reducing dimensionality of environmental variables

We obtained 11 temperature and nine precipitation bioclimatic variables at 30 s (~1 km2) resolution from the WorldClim 2.1 database, which includes climate data from 1970–2000 (Fick & Hijmans, 2017). Variables were centered, scaled, and reduced via principal components analyses (PCAs) using the prcomp() function in R, with separate analyses conducted for temperature and precipitation. The first two axes for temperature (hereafter, TPC1-2) explained a combined 88.7% of the variation and the first two axes for precipitation (PPC1-2) explained 95.8% of the variation (Fig. S2). Variable loadings indicated TPC1 primarily represented year-round temperatures, with higher values relating to warmer, more stable overall temperatures, while TPC2 represented mean diurnal temperature ranges, with higher values relating to more extreme daily temperature fluctuations. Similarly, PPC1 described the overall precipitation amount, with higher values indicating wetter conditions, whereas PPC2 represented precipitation variability, with higher values representing more variability in rainfall (Tables S5 and S6).

Identifying factors related to infection status and intensity via random forests

We employed the machine learning technique, RF, to identify important predictor variables related to infection status and infection intensity. We created classification models to predict pathogen status (infected or uninfected) for datasets with sufficient infection counts, including models encompassing all frogs (Bd_all: n = 1,281, Rv_all: n = 1,187) and family-specific models for Bd only (Bd_Bufonidae: n = 320, Bd_Hylidae: n = 456, Bd_Ranidae: n = 505). We also created a Bd infection status model excluding internal only and unknown tissues to investigate the impacts of tissue type on our results (Bd_external: n = 845). Additionally, we created an RF regression model for Bd infections (n = 216) to examine variables important to infection intensity (average SQ values). Models were created and assessed using the randomForests (Liaw & Wiener, 2002) and caret (Kuhn, 2008) R packages.

To address the large imbalance between infected and uninfected cases across the data, classification models were first balanced by either downsampling or upsampling. For downsampling, majority cases (uninfected) were randomly subsampled with replacement to equal the number of minority cases (infected). For upsampling, minority cases (infected) were randomly duplicated with replacement to equal the number of majority cases (uninfected). RF models were then optimized to predict both majority and minority cases by using the “strata” and “sampsize” functions to pull equal counts of infected and uninfected cases when creating forests. We assessed model overfitting via cross-validation and selected the model with the highest performance without overfitting, ultimately using downsampling for all models.

To further mitigate effects of small sample sizes of infected cases, we employed regularization via limited maximum depth by using the optimal mtry with the tuneRF() function of the randomForests package. We also used feature selection methods to reduce over-fitting of the data and determined variables for inclusion in our final classification models by comparing average out-of-bag (OOB) error rates. Cross-validations were performed using the predict() function from the stats package (R Core Team, 2023) to create a confusion matrix to assess predictive accuracy. To assess regression models, we used mean squared error and R² metrics. Final models were constructed with 2,000 trees and 100 permutations, runs were averaged over 100 iterations, and mean error rates were recorded. Average relative importance for each independent variable in the final model was measured using the importance() function and visualized using the ggplot2 package (Wickham, 2016).

Lastly, we evaluated final model performance, assessed predictive accuracy, and documented misclassification by splitting the data into training (70%) and testing (30%) datasets with equal proportions of positive infections and passed testing data through final models 100 times. We applied cross-validation to the final models to assess performance using the same methods described above.

One-way analyses of infection status and intensity

We used parametric and non-parametric tests via the stats R package to examine relationships between the potential factors associated with infection prevalence and intensity. We assessed model assumptions using the plotNormalhistogram() function in the rcompanion package and the var.test() function in the stats package. To assess differences in infection prevalence among categorical variables such as family, species, age class, sex, and tissue type, we used Pearson’s chi-squared tests for each pathogen type. We compared aspects of geography (latitude and elevation) and environmental variables (TPC1-2, PPC1-2) between infected and uninfected frogs using Student’s two-tailed t-tests. We performed logistic regressions to examine odds of infection across continuous variables such as latitude, longitude, elevation, TPC1-2, and PPC1-2 using the stats package and assessed model assumptions using the rcompanion package.

To examine differences in infection intensity between categories (family, species, age class, sex), we performed one-way ANOVAs assuming equal variances or Welch’s ANOVAs for tests with unequal variances. When comparing mean infection intensity between two groups (e.g., Rv species infection intensity, age class, and sex), we performed two-tailed Student’s t-tests. When significant differences were detected in ANOVA results, post-hoc analyses were carried out as follows. For parametric data, multiple two-tailed t-tests with Bonferroni correction or Tukey’s Honest Significant Difference tests were employed, while for non-parametric data, Games-Howell post-hoc tests were conducted using the rstatix package (Kassambara, 2023). Additionally, given the possibility of coinfections acting synergistically to increase negative infection outcomes (reviewed in Herczeg et al., 2021), we also evaluated differences in infection intensity between coinfected individuals and individuals with single infections of each pathogen using two-tailed t-tests. Finally, we performed linear regressions to assess infection intensity across continuous variables such as latitude, longitude, elevation, TPC1-2, and PPC1-2 using the stats package and assessed model assumptions using the erikmisc package (Erhardt, 2024). All R code and datasets are available from Figshare (https://doi.org/10.6084/m9.figshare.26849554), specimens are searchable via Arctos (arctos.database.museum; Cicero et al., 2024), and pathogen data are available on the Amphibian Disease Portal (Expedition GUID: https://n2t.net/ark:/21547/FkC2).

Results

Pathogen prevalence summary

Approximately 20% (n = 256) of the 1,281 individual frogs screened were infected with at least one pathogen, 231 with single infections and 25 with coinfections. We found substantial variation in prevalence among pathogens (Figs. 2 and S3), with Bd exhibiting the highest prevalence (16.9%; 95% CI [14.9–19%]), followed by Rv (4.38%; 95% CI [3.35–5.7%]) and Pr (1.06%; 95% CI [0.618–1.82%]). Significant variation in Bd and Rv infections was evident both between and within family groups (Fig. 2A; Table 1), with Pearson residual scores indicating higher-than-expected counts for both pathogens in the family Ranidae (Fig. S4). For Bd, we found higher counts of infection in the species R. catesbeiana, R. clamitans, and R. sphenocephala (Fig. S5). Within Bufonidae and Hylidae, Bd infections were primarily associated with a single species in each family, namely A. americanus and P. crucifer (Fig. 2A; Table S2). For Rv, we also observed higher-than-expected counts in Ranidae, specifically R. catesbeiana and R. clamitans (Table 1, Fig. S5). Pr prevalence was consistently low with no discernable pattern of infection across taxonomic groups.

Figure 2 Pathogen prevalence and intensity of infection across host taxonomy.

(A) The number (ni) of individuals infected out of the number sampled (n) and the associated proportion that tested positive for each pathogen (%). Phylogeny is based on relationships obtained from Vertlife.org (Jetz & Pyron, 2018) and edited using FigTree v1.3.1 (Rambaut, 2010). (B–E) Infection intensity values represented by log transformed starting quantities (SQ) for Bd and Rv pathogens with sample sizes in grey boxes. Significant differences (p < 0.05) between means for each paired comparison are indicated by differing letters (a, b). Silhouettes of frogs were obtained from phylopic.org.

Table 1 Summary of parametric and non-parametric statistical analyses.

	Bd status	Pr status	Rv status	
	p value	df	χ2	p value	df	χ2	p value	df	χ2	
Family	<0.001	2	134	0.499	2	1.4	<0.001	2	35.8	
Species	<0.001	10	179	0.922	6	1.98	<0.001	8	67.8	
Age class	0.134	1	2.25	–	–	–	<0.001	1	14.3	
Sex	0.918	1	0.011	0.58	1	0.308	0.599	1	0.278	
Tissue type	<0.001	1	59.7	0.724	1	0.125	0.874	1	0.025	
	p value	df	t stat	p value	df	t stat	p value	df	t stat	
Latitude	<0.001*	289	−4.76	0.005	13.2	3.35	0.002	1,185	−3.15	
Log10 (Elevation)	0.291	1,279	−1.06	0.687	1,222	−0.4	0.195	1,185	−1.3	
TPC1	<0.001*	289	5.29	0.038	12.9	−2.32	0.001	1,185	3.27	
TPC2	<0.001	290	4.79	<0.001	13	−6.29	0.354	1,185	0.928	
PPC1	0.211	1,279	1.26	0.259	12.9	−1.18	0.386	1,185	0.867	
PPC2	<0.001*	346	−3.36	0.641	1,222	0.466	0.131	1,185	−1.51	
	Bd intensity	Pr intensity	Rv intensity	
	p value	df	F/t stat	p value	df	F/t stat	p value	df	F/t stat	
Family	0.01	2, 189	(F) 4.69	0.351	2, 7	(F) 1.22	0.035	2, 2.62	(F) 15.5	
Species	0.002	5, 57.2	(F) 4.32	–	–	–	0.584	20	(t) 0.566	
Age class	0.004	156	(t) −3.63	–	–	–	0.807	21	(t) −0.25	
Sex	0.472	105	(t) −0.723	–	–	–	0.222	10	(t) 1.3	
Bd & Rv co-infection	0.589	181	(t) −0.541	–	–	–	0.931	27	(t) 0.088	
Bd & Pr co-infection	0.552	171	(t) 0.597	0.157	8	(t) 1.57	–	–	–	
Note:

Values from tests of pathogen status across 11 factors are shown at the top and tests of pathogen intensity across six factors are shown on the bottom. The test statistic (χ2, t stat, or F stat), degrees of freedom (df), and p value are shown for Bd, Rv, and Pr. Blank cells indicate tests were not run for that pathogen and factor combination because of insufficient data. Bold text denotes significance (p < 0.05). An asterisk (*) denotes variables that are no longer significant when excluding samples derived from internal and unknown tissue types.

Rv infection prevalence was notably higher in juveniles compared to adults (Tables 1 and S7). We found no significant difference in pathogen prevalence between sexes for any pathogen (Table S8). For Bd, as expected, there were significant differences based on tissue type, with fewer infections from internal only tissues compared to external and combination tissue types (Table 1; Fig S6; Table S9). When we re-ran Bd analyses excluding internal only tissues, family and species remained significant factors for Bd infection status (Table S10).

Bd coinfections represented more than half of the individuals infected with Pr (7 of 13, or 53%) and about a third of those infected with Rv (18 of 52, or 35%). Whereas only 12% of samples infected with Bd represented a coinfection (25 out of 216).

Infection intensity summary

Overall, infection intensities were low, with the highest values observed in Bd infections (Table S11). Bd and Rv intensity varied significantly among families, with the highest Bd intensity found in Ranidae, followed by Hylidae and Bufonidae (Table 1, Fig. 2B). For Bd, significant differences in infection intensity across families were primarily driven by Ranidae and Bufonidae, while for Rv, differences were driven by Hylidae and Ranidae (Figs. 2B and 2D; Table S11). No clear pattern was found in Pr infection intensities across families due to small sample sizes. Interspecific variation revealed differences in infection intensity among species, with significantly higher Bd intensity in R. clamitans compared to A. americanus, R. catesbeiana, and R. pipiens (Fig. 2C; Table S12). Rv intensity was somewhat higher in R. catesbeiana than R. clamitans, but the difference was not significant (Fig. 2E; Table S12).

We found slightly higher Bd infection intensity in juveniles compared to adults, but there was no significant variation in infection intensity for any pathogen screened related to sex (Fig. S7; Tables S13 and S14). When we re-ran Bd analyses excluding internal only tissues, family and species remained significant for Bd infection intensity, while age class became significant (Table S10). We found no significant differences between Bd infection intensity for individuals with single infections compared to individuals coinfected with Rv or Pr, nor were there differences in Rv or Pr infection intensity between single infections and coinfections with Bd (Table 1).

RF classification and regression models

The Bd_all RF classification model identified species-level taxonomic rank as the most important factor associated with Bd infection status (Fig. 3A). Overall annual temperature (TPC1), latitude, and daily temperature fluctuations (TPC2) were also important predictors (Fig. 3A). The most important variable differed depending on which infection status class was being considered, with species as the top predictor of uninfected Bd cases (Fig. 3B), while overall temperature (TPC1) and daily fluctuations (TPC2) showed greater accuracy in predicting infected Bd cases (Fig. 3C). The model exhibited a consistent error rate across classification types at 23.6%, achieving an overall accuracy of 74%. Notably, the model performed similarly at classifying both infected and uninfected cases with sensitivity and specificity scores of 0.70 and 0.79, respectively. The top predictors and model performance for Bd infection status were consistent when we excluded internal only tissues (Fig. S8).

Figure 3 Variable importance for random forests (RF) models of infection status.

(A) Overall model of Bd infection status. (B, C) Variables important for Bd uninfected and infected status, respectively. (D) Overall model of Rv infection status. (E, F) Variables important for Rv uninfected and infected status, respectively. For all panels, variable importance is measured as mean decrease in accuracy, averaged across 100 iterations, and ranked from highest (left) to lowest (right) as determined by balanced classification RF analyses. Average out-of-bag (OOB) classification error rates are shown. Predictive accuracy, sensitivity, and specificity values of the final models were derived from cross-validation, with 70% of the data used for training and 30% for testing.

Though there were fewer infection points to sample from, the Rv_all classification model also identified species as the most important predictor of Rv, followed by family, latitude, and TPC1 (Fig. 3D). Species was the most important predictor of uninfected status, while latitude and overall temperature (TPC1) were the top contributing factors to infected Rv cases (Figs. 3E and 3F). The model had an average error rate of 30.2%, with slightly lower error rates for predicting infected status (28.0%) compared to uninfected status (32.5%). Model validation showed moderate accuracy at 71%, with 0.75 sensitivity and 0.71 specificity.

Family-specific RF classification models highlighted differences in top predictors depending on the dataset (Fig. S9). Specifically, for the Bd_Bufonidae and Bd_Hylidae models, species was the most important factor (Figs. S9A–S9F). These results differed from the Bd_Ranidae model, where we observed that species had low importance for predicting infection status and that daily temperature fluctuation (TPC2) was instead the most important factor, followed by latitude and year-round temperature (TPC1; Figs. S9G–S9I). The Bd_Ranidae model had an average error rate of 27.7%, with slightly lower error rates at predicting infected status (27.4%) compared to uninfected (28.0%). Model validation showed moderate accuracy at 70%, with 0.62 sensitivity and 0.74 specificity. For Bd_Bufonidae and Bd_Hylidae, cross-validation revealed higher model accuracy (87% and 89%, respectively), but both models showed signs of model class imbalance, with low sensitivity values (0.5, and 0.55) and high specificity (0.9, 0.91; Figs. S9A and S9D).

The Bd infection intensity RF regression model indicated latitude, overall temperature (TPC1), and precipitation (PPC1) were the top predictors. The model only explained 12.6% of variation, however, with validation showing large mean absolute error (MAE = 1,173) and low R2 (0.14; Fig. S10).

Geographic and environmental predictors of infection

We observed significant differences in latitude between infected and uninfected individuals across all pathogens (Table 1). For Bd and Rv, infected individuals tended to occur at higher mean latitudes than uninfected (Figs. 4A and 4C), while Pr infections were only found at lower latitudes (Fig. 4B). Within species, the difference in mean latitude between Bd infected and uninfected samples was only statistically significant for R. clamitans, with infected samples tending to occur at higher latitudes (Fig. S11). A similar trend was observed in A. americanus and R. catesbeiana, while P. crucifer showed a trend in the opposite direction, with infected individuals tending to occur at lower latitudes.

Figure 4 Distribution of latitude and environmental variables across infected and uninfected groups.

(A, D, G, J) Individuals tested for Bd and found to be infected (teal) or uninfected (grey). (B, E, H, K) Individuals tested for Pr and found to be infected (tan) or uninfected (grey). (C, F, I, L) Individuals tested for Rv and found to be infected (purple) or uninfected (grey). Distribution of latitude in degrees (A–C), TPC1 (D–F), TPC2 (G–I), and PPC2 (J–L) for infected and uninfected individuals. Significant differences (p < 0.05) between means for each paired comparison are indicated by differing letters (a, b).

Elevation and overall precipitation (PPC1) did not exhibit significant differences between infected and uninfected individuals for any pathogen, but differences in the other environmental variables (TPC1, TPC2, PPC2) were significant for one or more of the three pathogens (Table 1; Fig. 4). For Bd and Rv, infected individuals were found at lower mean TPC1 than uninfected, suggesting an association with cooler year-round temperatures, while Pr infections were only found at higher TPC1 values (Figs. 4D–4F). For TPC2, Bd infections were associated with lower daily temperature fluctuations, Pr infections with higher daily temperature fluctuations, while the differences between infected and uninfected individuals were not significant for Rv (Figs. 4G–4I). PPC2 only differed for Bd, with slightly higher mean values for infected individuals compared to uninfected (Figs. 4J–4L), indicating Bd infections were associated with higher variability in rainfall. When we re-ran Bd analyses excluding internal only tissues, only differences in TPC2 remained significant (Table S10).

No significant relationships were identified between infection intensity and geographic and environmental factors for any pathogen, but there was a trend of higher infection intensities associated with lower TPC2, which only increased in significance once we removed internal-only samples.

Discussion

Our study design enabled comparisons of infection dynamics across multiple pathogens infecting widespread, co-distributed host taxa, while accounting for geographic and environmental variation. Through this design, we detected different infection patterns across pathogens, temperature conditions, host taxonomy, and age groups of frogs distributed throughout the central and eastern United States. Even though our study did not sample across seasons, which are known to impact amphibian pathogen dynamics (Duffus et al., 2015; Sasso, McCallum & Grogan, 2021; Atkinson & Savage, 2023b), we were able to capture significant prevalence of amphibian pathogens across this region. We observed widespread Bd infections at low relative intensities across 29 states, supporting the idea that this pathogen is endemic throughout much of this region. We also found that coinfections accounted for more than half of Pr and a third of Rv infections, despite these pathogens typically affecting only larval stages. This suggests that coinfections in adults may play a role in altering susceptibility to pathogens that usually target earlier life stages. In our dataset, individuals infected with Bd and Rv were associated with higher latitudes with lower, more stable overall temperatures, while Pr infections were limited to southern latitudes with increased daily temperature fluctuations, though we note that additional Pr sampling is needed.

Our balanced RF models successfully identified factors important to Bd and Rv infection status. Specifically, each model determined host species as the top predictor with ~70% accuracy, suggesting other factors not captured here are also important in predicting infection status. Broadly speaking, species in the family Ranidae, especially Rana catesbeiana and R. clamitans, harbored the majority of Bd and Rv infections, adding to the mounting evidence that these species act as potential pathogen reservoirs in this region. Furthermore, we detected higher Rv prevalence and Bd infection intensity in juvenile frogs compared to adults. These findings underscore the complex interplay between host traits, environmental conditions, and pathogen dynamics in shaping infection patterns across this region. We examine these results in relation to previous amphibian pathogen research, discuss sampling limitations, and highlight the challenges and advances that machine learning can provide to large-scale wildlife disease research in the sections below.

Prevalence and geographic distribution vary among pathogens

Batrachochytrium dendrobatidis (Bd)—Our survey is consistent with previous reports (Li et al., 2021), distribution models (Rödder, Schulte & Toledo, 2013; Olson et al., 2021), and public database counts (https://amphibiandisease.org; Koo et al., 2021) that indicate the widespread occurrence of Bd across much of the central and eastern United States. We found at least one frog infected with Bd in 29 of the 32 states screened, reinforcing the conclusion that Bd is likely endemic throughout much of this region (Longcore et al., 2007). We also note that northern states such as Maine, Vermont, and Indiana exhibited the highest proportion of Bd infections, with nearly half of all frogs screened testing positive. This high relative prevalence at higher latitudes could be explained by lower overall summer temperatures, which support fungal growth while limiting exposure to above-optimal temperatures that slow disease progression (Lindauer, Maler & Voyles, 2020). Similar patterns have been observed across the U.S. (Petersen et al., 2016; Sonn, Utz & Richards-Zawacki, 2019; Belasen et al., 2024) and Australia (Kriger, Pereoglou & Hero, 2007). However, it is important to acknowledge that our assessment likely underestimates the prevalence of Bd within southeastern regions since many of these samples were derived from internal tissues (e.g., liver, muscle) which are not reliable when screening for Bd (Fig. S1; World Organisation for Animal Health (WOAH), 2021a).

When we excluded samples derived from internal and unknown tissue types, TPC2 (corresponding to mean diurnal temperature range) remained significantly correlated with Bd prevalence. Experimental studies have shown reduced growth and reproduction in Bd when exposed to heat pulses compared to constant temperature treatments (Greenspan et al., 2017; Lindauer, Maler & Voyles, 2020). Likewise, our results suggest that areas with more stable daily temperatures—such as humid, temperate regions near bodies of water—may serve as important refugia for Bd, while more fluctuations in daily temperatures result in decreased pathogen prevalence. Although extensive research has examined climate in relation to Bd infection patterns (e.g., reviewed in Sasso, McCallum & Grogan, 2021), the effect of daily temperature fluctuations in relation to pathogen prevalence and distribution is underreported.

Amphibian Perkinsea (Pr)—Our results support three key insights regarding the distribution of Pr. First, our study suggests the current known distribution of Pr is likely underestimated, as we document novel infections in Missouri and Oklahoma (Isidoro-Ayza et al., 2017; Garner & Ruden, 2019; Midwest Association of Fish and Wildlife Agencies (MAFWA), 2020). Second, the westward range expansion to these states, along with reported mortality events as far north as Wisconsin (Isidoro-Ayza, 2019) and Alaska (Isidoro-Ayza et al., 2017), indicate that Pr likely possesses broad climatic tolerances. Although researchers have yet to successfully culture Pr in the lab to explore these limits, previous surveys have identified seasonal Pr outbreaks throughout the southeastern U.S. (Atkinson & Savage, 2023b). Lastly, the low infection prevalence we document here supports the hypothesis that Pr infections are currently localized and rare in adult frogs during summer months, though not impossible (Jones et al., 2012; Karwacki et al., 2018). Nonetheless, with rising concern for species in recent decline, potentially due to the introduction of Pr, such as Rana capito (Crawford et al., 2022; Devitt et al., 2023) and Rana sevosa (Atkinson, 2016), we recommend targeted screening across seasons of abundant co-distributed adult frogs that do not currently show significant population declines, such as R. catesbeiana, R. sphenocephala, and R. clamitans, to assess how these pathogens are potentially spread and maintained in the wild.

Ranavirus (Rv)—Rv has been documented widely across North America, as shown in the Global Ranavirus Reporting System (brunnerlab.shinyapps.io/GRRS_Interactive/) and subsequent reviews (e.g., Duffus et al., 2015; Brunner et al., 2021). It is important to note that the GRRS database includes reports of infections in both wild and captive frog populations. Therefore, even though Rv distributions are recorded widely, some states only have Rv outbreaks documented in captivity (Duffus et al., 2015). Our study adds such a novel case of Rv infection in wild frogs of Mississippi. The majority of Rv infections, however, were detected in northern states such as Maine and New Jersey, where 28% and 13% of individuals tested were found positive, respectively. Similar to Bd research, and in line with previous observational studies (e.g., Youker-Smith et al., 2018), we observed significantly higher Rv prevalence in areas characterized by cooler year-round temperatures. We note, however, that the higher Rv prevalence we found in high-latitude areas may be influenced by the species sampled at northern sites (primarily Rana catesbeiana and R. clamitans) and the age class of those sampled (many juveniles), both of which are factors that could conceivably correspond with increased infection.

The relatively low overall prevalence of Pr and Rv in our study was likely influenced by a few factors. First, while both liver and toe clips can reliably detect Rv (Torres López et al., 2024) and are recommended diagnostic tissues (World Organisation for Animal Health (WOAH), 2021b), the use of toe clips for Pr detection has not yet been validated. Additionally, the absence of larval stages in our sampling likely had a strong impact on prevalence of both Pr and Rv. While infections (Gray, Miller & Hoverman, 2009; Karwacki et al., 2018) and mortality (Teacher, Cunningham & Garner, 2010) have been documented in adult frogs, mass die-off events for both pathogens primarily occur before and during metamorphosis (Rv: Green, Converse & Schrader, 2002; Pr: Isidoro-Ayza, 2019). This trend is at least partly explained by the underdeveloped immune response documented in hosts prior to and during larval development (Gray, Miller & Hoverman, 2009; Miller, Gray & Storfer, 2011; Herczeg et al., 2021), as well as the habitat overlap between these water-borne pathogens and larval stages (Miller, Gray & Storfer, 2011; Itoïz et al., 2022). We did, however, record higher than expected Rv infections in post-metamorphic juvenile frogs compared to adults, perhaps explained by the continued lag in host immune function between metamorphosis and sexual maturity (Rollins-Smith, 1998; Gantress et al., 2003). Lastly, the presence of both Rv and Pr infections in adult frogs in our study may either represent coincidental infections prior to clearance or be related to other mechanisms, such as coinfections, discussed below.

Coinfections account for many cases of rare pathogen infections

Even though we report no significant differences in infection intensity between single and coinfected individuals, we still document a surprising proportion of coinfections between Bd-Pr and Bd-Rv. This observation, along with findings from other recent multi-pathogen screening efforts (e.g., Landsberg et al., 2013; Karwacki, Martin & Savage, 2021; Atkinson & Savage, 2023b), support the idea that coinfections may facilitate heightened susceptibility in frogs (Herczeg et al., 2021), but more work is needed to understand how these coinfections are affecting seemingly robust species such as Rana catesbeiana and R. clamitans. Though we did not find coinfections between Pr and Rv, which primarily impact larval stages, infection with Bd may increase susceptibility of adult frogs to Pr and Rv. One concerning aspect of this finding is that given their increased immune function, adults can maintain chronic, low-level infections, thus providing a potential mechanism for pathogen spread and persistence in populations. Further investigation in a controlled setting is warranted to assess how coinfections at different life stages may impact the infection dynamics and outcomes we observe in natural environments and communities.

Infection prevalence differs across host taxonomy

Our study adds to a growing body of research that aims to understand differences in infection prevalence among hosts. Specifically, we found significantly higher Bd and Rv prevalence in four species in the family Ranidae compared to other families (Fig. 2; Fig. S2), while Pr counts were too low to draw significant conclusions. Higher pathogen prevalence in the family Ranidae has been reported from other surveys of wild communities (e.g., Olson et al., 2013; Rothermel et al., 2016; Karwacki et al., 2018), and increased susceptibility has been directly reported in controlled experiments (e.g., Hoverman, Gray & Miller, 2010; Gahl, Longcore & Houlahan, 2012). One potential explanation for increased pathogen prevalence in this family could be proximity and time spent in permanent and semi-permanent breeding ponds that act as pathogen refugia (Hoverman et al., 2011; Greenberg, Palen & Mooers, 2017).

Two species, Rana catesbeiana and R. clamitans, accounted for 38% of all infected frogs and 64% of all coinfections in our dataset. Both species have been classified as pathogen reservoirs which contribute to the maintenance of Bd and Rv across their native ranges over time (Rothermel et al., 2016; Yap et al., 2018; Brunner et al., 2019; Hoverman, Chislock & Gannon, 2019; Hossack et al., 2023). In addition, the global invasion of pathogen-positive R. catesbeiana has been linked to the spread of both Bd and Rv with primarily negative effects for native species (reviewed in Atkinson & Savage, 2023a). Other species-level differences observed in our study, such as increased Bd prevalence in Anaxyrus americanus and Pseudacris crucifer compared to other species in their respective families, also warrants further investigation to identify potential drivers of increased susceptibility outside of primary habitat type. Research into mechanisms of resistance and tolerance (e.g., Eskew et al., 2015), host-pathogen coevolutionary history (e.g., Carvalho et al., 2024) and immunogenetic adaptation (e.g., Trujillo et al., 2021) focused on additional host species could provide further important insights.

Infection intensity varies across host taxonomy and age class

In amphibians, infection intensity serves as a proxy for infection outcomes, with higher pathogen loads often corresponding with increased spread and host mortality (Vredenburg et al., 2010). Our survey of widespread species that are not currently documented as experiencing pathogen-induced decline, still showed significant associations between infection intensity, host species, and age class. Specifically, we found the highest Bd infection intensities in Rana clamitans (Fig. 2B) and significantly higher Bd intensity in juveniles compared to adults (Table S1). The majority of Bd-positive juvenile frogs in our dataset were R. clamitans, however, meaning that our data lack the power to distinguish infection intensity variation between age classes and species-level correlates. Importantly, multiple studies have shown that Bd infection intensities increase during winter months (Savage, Sredl & Zamudio, 2011), which our study also fails to capture. Therefore, more research is needed to assess the differences in Bd intensity across life stages within R. clamitans and across seasons to extrapolate the mechanisms responsible for this pattern.

Broadly, age class has been associated with pathogen-induced host mortality, but these results have been mixed. For example, Bd-induced mortality has been documented during metamorphosis (Humphries et al., 2024) and in juvenile stages (Abu Bakar et al., 2016), but also the inverse has been recorded, where juveniles harbor lower infection loads compared to adult frogs (Bradley et al., 2019). This inconsistency demonstrates the contextual nature of infection dynamics, where multiple aspects can influence infection outcomes. Factors such as length of exposure (Lips, 2016; Bielby et al., 2021), host skin peptides and microbiomes (Woodhams et al., 2007; Bernardo-Cravo et al., 2020), infection history (Greenberg, Palen & Mooers, 2017), and phylogenetic constraint (Hoverman et al., 2011; Longo, Lips & Zamudio, 2023) have all been shown to affect pathogen-specific infection intensities and/or host mortality. Continued work documenting species-specific mechanisms of pathogen tolerance and resistance is needed to better understand the drivers of infection intensity differences and host survival.

Application and limitations of RF models in multi-pathogen dynamics

Our application of balanced RF classification models builds on efforts that have used machine learning to address factors influencing both species-level (Murray et al., 2013) and site-specific (Atkinson & Savage, 2023b; Roth, Griffis-Kyle & Barnes, 2024) amphibian infection dynamics. Machine learning techniques have also been applied to examine projected distributions of Bd in the context of future climate change (Xie, Olson & Blaustein, 2016). We demonstrated the utility of RF models to quickly and efficiently provide valuable insights into the complex patterns encountered in observational disease ecology studies (Cutler et al., 2007). Though there are limitations such as over-fitting due to small sample sizes or amplifying noise in the data rather than true relationships, models built from balanced training datasets with ample sample sizes can achieve moderately low out-of-bag error rates and have high predictive accuracy, specificity, and sensitivity. The application of RF models to other wildlife disease systems such as avian malaria-causing parasites (e.g., Aželytė et al., 2023), Pseudogymnoascus destructans, which causes white-nose syndrome in bats, or Ophidiomyces ophiodiicola, which causes snake fungal disease, are other promising avenues for discovery.

Conclusions

Frogs in North America have faced infectious disease pressures from emerging pathogens for over a century (Karwacki, Martin & Savage, 2021). While fine-scale, pathogen-specific research has been essential for understanding the mechanisms driving large-scale infection dynamics, our study underscores the need for multi-pathogen screening efforts to effectively monitor diseases across diverse frog communities. We showed that host species and environmental factors were top predictors of pathogen prevalence, but their relative importance differed among pathogens and host families. Moreover, large-scale studies like ours depend on tissue donations to natural history collections. This practice, along with the deposition of whole specimens, can provide valuable histological and morphometric insights that extend beyond the sample’s original research scope, promoting reproducible and extendable science. Lastly, amphibian pathogens present a unique opportunity to develop machine learning models that capture multi-pathogen infection dynamics on a broad scale. The extensive body of research available within this field can enhance model predictions, while the substantial screening efforts can provide ample data to yield valuable insights, when made available via public databases. Overall, the multifaceted nature of host-pathogen dynamics poses a challenge to comprehensive, large-scale wildlife disease studies; therefore, future research that embraces data sharing and flexible analytical approaches will pave the way for deeper insights into these complex systems.

Supplemental Information

Supplemental Information 1 Sample information.

Supplemental Information 2 Pathogen prevalence and intensity summaries and associated statistical analyses.

Supplemental Information 3 Distributions of tissue types used in DNA extractions by species.

Individual counts for tissue type extracted and used for Bd screening are shown for each species. Stars denote species found in the southeastern U.S. region that were sampled prior to 2021. Combination tissues include both internal (muscle, liver) and external (toe/tail, including skin) tissue.

Supplemental Information 4 First two principal component axes for temperature and precipitation across family.

Individual points are colored by family and represent values along the first and second PC dimensions which represent 11 bioclimatic variables relating to temperature (left) and 9 bioclimatic variables relating to precipitation (right).

Supplemental Information 5 Pathogen prevalence (proportion of individuals infected) across sampling distribution.

Pie charts represent the proportion of infected individuals pooled within a 50 km radius for (a, d) Bd, (b, e) Pr, and (c, f) Rv. Uninfected sites are pooled within a 100 km radius and circle sizes reflect the number of individuals at that site. (d–f) Proportion of infected individuals at the community level, pooled at <50 km for our densest sampling site (FL2) with number of individuals below.

Supplemental Information 6 Bd and Rv infection prevalence by family.

Correlation plots show the direction and contribution to Pearson’s residuals of Bd (left) and Rv (right) infection counts by family. Positive residuals are blue, suggesting a positive association between the corresponding row and column, and negative residuals are red, suggesting a negative association. Size of the circle indicates relative contribution to Pearson’s residual.

Supplemental Information 7 Bd and Rv infection prevalence by species.

Correlation plots show the direction and contribution to Pearson’s residuals of Bd (left) and Rv (right) infection counts by species. Positive residuals are blue, suggesting a positive association between the corresponding row and column, and negative residuals are red, suggesting a negative association. Size of the circle indicates relative contribution to Pearson’s residual.

Supplemental Information 8 Bd infection prevalence by tissue type.

Correlation plots show the direction and contribution to Pearson’s residuals of Bd infection counts by tissue types used in DNA extraction. Positive residuals are blue, suggesting a positive association between the corresponding row and column, and negative residuals are red, suggesting a negative association. Size of the circle indicates relative contribution to Pearson’s residual.

Supplemental Information 9 Distribution of average infection intensities between age and sex for each pathogen.

Infection intensity is measured in log(SQ) across (a) age classes and (b) sex for Bd (left), Pr (center), and Rv (right) infections. Significant differences (p < 0.05) between means for each paired comparison are indicated by differing letters (a, b).

Supplemental Information 10 Variable importance for random forests (RF) classification models of infection status built excluding internal only tissues.

(a) Overall model of Bd infection status. (b-c) Variables ranked by importance for Bd uninfected and infected status, respectively. For all panels, variable importance is measured as mean decrease in accuracy, averaged across 100 iterations, and ranked from highest (left) to lowest (right) as determined by balanced classification RF analyses. Average out-of-bag (OOB) classification error rates are shown. Predictive accuracy, sensitivity, and specificity values of the final models were derived from cross-validation, with 70% of the data used for training and 30% for testing.

Supplemental Information 11 Variable importance for random forests (RF) classification models of Bd infection status by family.

(a) Overall model of Bd infection status for Bd_Bufonidae models only. (b, c) Variables ranked by importance for Bd uninfected and infected status, respectively. (d) Overall model of Bd infection status for Bd_Hylidae models only. (e, f) Variables important for Bd uninfected and infected status, respectively. (g) Overall model of Bd infection status for Bd_Ranidae models only. (h, i) Variables important for Bd uninfected and infected status, respectively. For all panels, variable importance is measured as mean decrease in accuracy, averaged across 100 iterations, and ranked from highest (left) to lowest (right) as determined by balanced classification RF analyses. Average out-of-bag (OOB) classification error rates are shown. Predictive accuracy, sensitivity, and specificity values of the final models were derived from cross-validation, with 70% of the data used for training and 30% for testing.

Supplemental Information 12 Random forest (RF) regression output for infection intensity of Bd.

Variable importance is measured as percentage mean squared error increase decrease, averaged across 100 iterations, and ranked from highest (left) to lowest (right) as determined by RF regression analysis. Percent variance explained is shown. R2 and mean absolute error (MAE) were derived from model validation, with 70% of the data used for training and 30% for testing.

Supplemental Information 13 Distribution of latitude between infected and uninfected species.

(a) Species with adequate number of Bd infections (n > 15 positive individuals per species) and (b) species with adequate number of Rv infections (n > 2 positive individuals per species). Significant differences (p < 0.05) between means for each paired comparison are indicated by differing letters (a, b).

Supplemental Information 14 ARRIVE 2.0 Checklist.

This research is part of the dissertation of DLFW. The authors thank the members of the Amphibian and Reptile Biodiversity Lab for their valuable input on earlier versions of the manuscript and analyses; J. Tom Giermakowski and members of the Division of Amphibians and Reptiles and Mariel Campbell of the Division of Genomic Resources, and the LSU Museum of Natural Science Collection of Genetic Resources, for access to materials and tissues; Melissa Sanchez for laboratory assistance and equipment; and Erik Erhardt in the Department of Mathematics and Statistics and Davorka Gulisija in the Department of Biology for consultation on statistical analyses. We also thank the creators of the original silhouette images: Steven Traver (Bufo gargarizans gargarizans), Jose Carlos Arenas-Monroy (Sarcohyla bistincta), and Beth Reinke (Rana temporaria).

Additional Information and Declarations

Competing Interests

The authors declare that they have no competing interests.

Author Contributions

Daniele L. F. Wiley conceived and designed the experiments, performed the experiments, analyzed the data, prepared figures and/or tables, authored or reviewed drafts of the article, and approved the final draft.

Kadie N. Omlor performed the experiments, authored or reviewed drafts of the article, and approved the final draft.

Ariadna S. Torres López performed the experiments, authored or reviewed drafts of the article, and approved the final draft.

Celina M. Eberle performed the experiments, authored or reviewed drafts of the article, and approved the final draft.

Anna E. Savage conceived and designed the experiments, authored or reviewed drafts of the article, and approved the final draft.

Matthew S. Atkinson conceived and designed the experiments, authored or reviewed drafts of the article, and approved the final draft.

Lisa N. Barrow conceived and designed the experiments, authored or reviewed drafts of the article, and approved the final draft.

Animal Ethics

The following information was supplied relating to ethical approvals (i.e., approving body and any reference numbers):

Florida State University Animal Care and Use Committee; University of New Mexico Institutional Animal Care and Use Committee.

Field Study Permissions

The following information was supplied relating to field study approvals (i.e., approving body and any reference numbers):

Alabama Department of Conservation and Natural Resources; Arkansas Game and Fish Commission; Florida Fish and Wildlife Conservation Commission; Georgia Department of Natural Resources; Iowa Department of Natural Resources; Illinois Department of Natural Resources; Indiana Department of Natural Resources; Kansas Department of Wildlife and Parks; Kentucky Department of Fish and Wildlife Resources; Louisiana Department of Wildlife and Fisheries; Maryland Department of Natural Resources; Maine Department of Inland Fisheries and Wildlife; Michigan Department of Natural Resources; Minnesota Department of Natural Resources; Missouri Department of Conservation; Mississippi Department of Wildlife, Fisheries, and Parks; North Carolina Wildlife Resources Commission; North Dakota Game and Fish Department; Nebraska Game and Parks Commission; New Jersey Division of Fish and Wildlife; New Mexico Department of Game and Fish; New York State Department of Environmental Conservation; Ohio Department of Natural Resources; Oklahoma Department of Wildlife Conservation; Pennsylvania Fish and Boat Commission; South Carolina Department of Natural Resources; State of South Dakota Department of Game, Fish, and Parks; Tennessee Wildlife Resources Agency; Texas Parks and Wildlife; Virginia Department of Wildlife Resources; Vermont Fish and Wildlife Department; Wisconsin Department of Natural Resources.

Data Availability

The following information was supplied regarding data availability:

The data and r-scripts are available at Figshare: Wiley, Daniele (Dani) (2024). CFPCS DATA & R-SCRIPTS. figshare. Dataset. https://doi.org/10.6084/m9.figshare.26849554.v5.

The pathogen screening data are available at the Amphibian Disease Portal (GEOME): https://n2t.net/ark:/21547/R2587.

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
