# Peer review of "Leveraging machine learning to uncover multi-pathogen infection dynamics across co-distributed frog families"

_PeerJ, doi:10.7717/peerj.18901_

## Round 0.1 · original submission · Minor Revisions

I have received reviews from two individuals who both think that this study is impressive and will make a solid contribution to the field. The reviewers, however, did point out a few minor issues that the authors should aim to rectify in their revisions. Specifically, caution should be offered with respect to an overreliance on qPCR results for determining infection status and, importantly, additional clarity is needed on how the origin of samples (internal vs. external) bear on false negative outcomes. Other suggestions for improvement are offered, especially related to the extent that your results and methods are in lock-step with your claims about their importance. The authors should consider these additional comments in their revisions. Thank you for you submission to PeerJ.

Reviewer 1 ·

Basic reporting

I am impressed by the work carried out by this team of researchers, and am happy to accept the manuscript as is. The researchers have provided sufficient background information, there are no additional references that came to mind when reading the manuscript. The structure is impeccable and the authors have clearly taken some time to think about the layout and the reader. All of the figures are clear and the research is self-contained.

Experimental design

I am happy with the scope of the experimental design, which also fits within the scope of the journal. The researchers have undergone rigorous investigation and performed some extraordinary work which will no doubt continue to benefit the conservation of amphibians as artificial intelligence and related technologies continue to grow in popularity and advancement. The methods are also described with sufficient detail that I hope to be able to replicate them in the future, given my current understanding of this technology. I would caution the authors though about relying solely on qPCR results to determine infection status in wild amphibians and suggest that in the future they work with veterinary staff to also help determine the impacts of infections/co-infection on individual frog health where possible.

Validity of the findings

The findings of this piece of research are valid, even if some of the results could be a little more robust - however this is one of the limitations of this kind of study. The conclusions and their implications are well stated, linking back to the original research question. Thankfully, the authors have not gone off track as so often happens and have kept their findings succinct and relevant.

Additional comments

This was a joy to read and it is very clear that sometimes ambition pays off. Well done everyone.

Reviewer 2 ·

Basic reporting

This paper uses widespread sampling of tissue samples from 12 widespread species in three families to assess the distribution of infections by three pathogens and to discover abiotic factors correlated with infections and co-infections in this amphibian community of the eastern United States.

The manuscript is well written and clearly outlines methods and results. The broader implications of the study are well set out in the Introduction.

The random forest analyses are a new approach in the context of looking at correlates of disease, and it is emphasized in the title of the manuscript and in the Intro. However, the RF analysis is only one small piece of the overall analyses, so it felt a bit oversold. I would suggest toning down that aspect.

Data availability is clearly stated.

Experimental design

The focus on multiple host species and three pathogens of frogs is novel, especially when considering the geographic scale of the study. Using random forest models also is novel and helps disentangle possible factors contributing to the distribution of pathogen infections.

Sample sizes were large, appropriate for the balanced RF analyses, to account for the large number of uninfected individuals typical of disease surveys. The screening design (with two negative replicates considered uninfected, and re-screening of positives for a total of four screens per individual, is sufficiently thorough.

One point that could be clarified is the distribution of internal/external samples (lines 197-203). As the authors point out, internal tissues are not expected to reliably diagnose Bd infections. It sounds from this description that that samples were included anyway, despite the high probability of false negatives. I see that later in the MS the authors state that they test the sensitivity of findings with those samples excluded for the Bd infection numbers. Likewise, what evidence do we have for diagnostic accuracy when testing for Ranavirus from internal and external samples? I realize that these tissues were collected from museum tissue collections (as well as from the field) so the availability of tissues (internal and external) will depend on the availability of those samples deposited in collections. A perfect experimental design would be to have both internal and external samples for every individual included in the study, so that prevalence and infection could be standardized for the tissue most accurate for diagnosis of each pathogen. Given that is not the case here, it might be important to validate the role that false negatives (based on tissue types) might have on the patterns detected.

The authors mention the possibility of co-infections increasing negative infection outcomes (line 315). Do you mean that coinfection could act antagonistically, and therefore reduce the intensity of one of the pathogens? This could be stated more clearly, citing the literature on antagonistic vs synergistic outcomes of coinfections. These analyses could also be done with prevalence, in addition to intensity. In cases of antagonistic pathogen interactions, we might expect species with one pathogen to have reduced infections of other pathogens.

Validity of the findings

The results largely mirror what we know about pathogen performances in different environments (Bd and Rv more prevalent at higher latitudes, Pr at lower latitudes). The study does add some novel findings for differences in pathogen infections among species, age classes, and specific combinations of abiotic factors.

Line 440. Here you suggest that Bd infections may facilitate infections by Rv and Pr. I am not sure your data actually support this. First, I don’t see how these data show the directionality of synergistic pathogen interactions. It could just as easily be that the other two pathogens facilitate Bd infections. Second, your coinfection cases are actually quite low, so this claim of synergistic infection dynamics seems to be a bit of a stretch.

Figure 4 would be easier to interpret if pathogen names were included along the side.

---

## Round 0.2 · accepted · Accept

Thank you for addressing the concerns in your revised submission. I did not feel that another round of reviews was necessary. I am happy to accept the current version. Thank you for your submission to PeerJ and congratulations on a great study!